# Enhancing Transformer-based Semantic Matching for Few-shot Learning through Weakly Contrastive Pre-training

## ABSTRACT

The task of semantic text matching focuses on measuring the semantic similarity between two distinct texts and is widely applied in search and ranking scenarios. In recent years, pre-trained models based on the Transformer architecture have demonstrated powerful semantic representation capabilities and have become the mainstream method for text representation. The pipeline of fine-tuning pre-trained language models on downstream semantic matching tasks has achieved promising results and widespread adoption. However, practical downstream scenarios often face severe challenges in terms of data quality and quantity. Ensuring high-quality and large quantities of samples is often difficult. Current research on enhancing pre-trained models for few-shot semantic text matching tasks is still not advanced enough. Therefore, this paper focuses on providing a general enhancement scheme for few-shot semantic text matching tasks. Specifically, we propose an Enhanced Transformer-based Semantic Matching method for few-shot learning through weakly contrastive pre-training, which is named as EBSIM. Firstly, considering the characteristics of semantic text matching tasks, we design a simple and cost-effective data augmentation method for constructing weakly supervised samples. Then, we design a contrastive learning objective based on alignment-aspect to achieve effective semantic matching by optimizing the bidirectional semantic perception between constructed texts. We conduct comprehensive experiments on five Chinese and English semantic text matching datasets using various Transformer-based pre-trained models. The experimental results confirm that our proposed method significantly improves the model's performance on semantic text matching tasks. Further ablation experiments and case studies validate the effectiveness of our approach. Our code and data will be made publicly available at a later stage.

## KEYWORDS

Text Semantic Matching, Pre-trained Language Model, Contrastive Learning

### ACM Reference Format:

Anonymous Author(s). 2024. Enhancing Transformer-based Semantic Matching for Few-shot Learning through Weakly Contrastive Pre-training. In *MM '24: ACM International Conference on Multimedia, October 28 - November 01, 2024, Australia.* ACM, New York, NY, USA, 10 pages. https://doi.org/10.1145/nnnnnnn.nnnnnnn

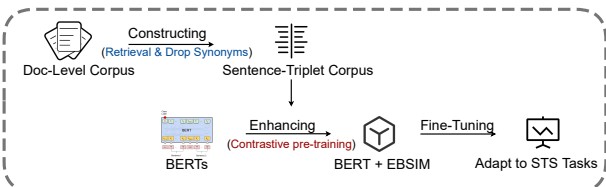

**Figure 1: Illustration of the EBSIM's pipeline. Firstly, the sentence-triplet corpus is constructed from a large-scale document-level corpus via token level and structure level data augmentation. Then, the BERTs are further pre-trained through contrastive objectives on the weakly-supervised corpus. Finally, the enhanced BERTs adapt to STS tasks in the fine-tuning stage. The subgraph of the BERT's structure is derived from Devlin et al. [16].**

## 1 INTRODUCTION

The semantic text similarity (STS) [20, 31, 40] task measures the semantic similarity between texts. It is a fundamental task in natural language processing (NLP). And it plays an essential role in various fields, such as information retrieval and machine translation evaluation. Plenty of methods have been proposed for the task, including traditional feature engineering techniques, hybrid approaches, and purely neural architectures [6, 33, 46, 54, 61].

Recently, pre-trained language models (PLMs) have led to impressive performance gains across various NLP tasks, such as sentiment classification [18, 52], natural language inference [10, 25], named entity recognition [27, 43] and so on. In detail, people generally load the PLM such as BERT [16], RoBERTa [38], then fine-tune it on the specific downstream task to obtain better performances. In this way, PLMs can be fine-tuned with just one additional output layer to create state-of-the-art models.

Despite the numerous research efforts that have introduced various pre-trained language models (PLMs) to enhance the performance of semantic matching tasks, there remain several critical issues that have not been effectively addressed. Firstly, there exists a natural gap between the learning objectives during the pre-training stage and those during the fine-tuning stage. The pre-training phase primarily focuses on language modeling aspects, such as the masked language model (MLM) in BERT, dynamic masking in RoBERTa, or sentence ordering relations modeling, exemplified by tasks like next sentence prediction (NSP) in BERT and sentence order prediction (SOP) in ALBERT [32]. However, semantic matching tasks during the fine-tuning stage are primarily concerned with learning semantic similarity between texts. The significant differences in learning objectives can easily constrain the performance of PLMs on downstream tasks. Secondly, obtaining high-quality and high-quantity downstream task datasets for fine-tuning presents a significant

challenge. The most critical factor affecting fine-tuning model effectiveness is the quality of training samples. Limited, noisy, and low-quality training samples can hinder model convergence and potentially disrupt the general language knowledge acquired during pre-training. The question of how to design effective pre-trained language models to endow them with strong text representation capabilities in few-shot learning scenarios is a pivotal issue.

In addition, semantic matching tasks place a greater emphasis on a model's ability to judge the semantic similarity between texts, as opposed to textual similarity judgments. Currently, there is limited research focused on designing pre-training models that are more tailored to downstream semantic matching tasks. To enhance a model's ability to recognize semantic consistency in textual content, a recent study introduced an approach involving the inclusion of sentiment recognition auxiliary tasks during the pre-training phase. Given the ubiquity and importance of semantic matching tasks, the development of pre-training methods specifically tailored to such tasks is a significant and valuable endeavor. Furthermore, when confronted with situations where two texts share the same synonyms, models can easily make inaccurate semantic judgments due to the presence of identical tokens. Therefore, devising effective pre-training strategies to mitigate a model's overreliance on synonyms and enhance its capacity for learning semantic structures is also a crucial consideration.

To address the aforementioned issues, this paper focuses on providing a general enhancement pre-training approach tailored to semantic text matching tasks with limited samples. Specifically, we propose an Enhanced Transformer-based Semantic Matching method for Few-shot Learning through Weakly Contrastive Pre-training, which is named as EBSIM. Firstly, to improve the alignment between the learning objectives in the pre-training and fine-tuning stages, we design a weakly supervised dataset for semantic text matching tasks based on general corpora. Considering the need to align the enhanced samples and training objectives with the characteristics of semantic text matching, we devise a triplet data augmentation method based on both text token-level and sentence structure-level. Particularly, we leverage high-quality article headlines from general corpora, such as Wikipedia, as anchors. We select relevant samples from the articles based on the similarity at the token level to construct pairs of samples. Given that models can be easily perturbed by similar sentence structures in matching tasks, we adopt a sentence structure-level approach by masking synonymous entity words to preserve the sentence structures in the text. This process helps construct pairs of samples.

Next, to effectively leverage the information from the weakly supervised dataset for consistent learning between pre-training and fine-tuning, we introduce a multi-aspect contrastive pre-training enhancement approach. We approach this from a relation-aspect perspective, utilizing the global semantic space obtained through Transformer encoding of text. We construct contrastive learning objectives based on the semantic representations of text pairs. Subsequently, we design contrastive learning objectives based on Alignment-Aspect, facilitating bidirectional semantic alignment between texts to achieve effective semantic matching. By enhancing Transformer-based pre-training models within this adaptable framework, we achieve improved performance in semantic text matching tasks. During the fine-tuning stage, there is no need to introduce additional network structures or modify pre-trained models; training and predictions can be directly performed based on downstream datasets. The whole pipeline of EBSIM is given in Figure 1.

The major contributions can be summarized as follows:

- We have designed a low cost and efficient Weakly-Supervised Data Augmentation method. From both text token-level and sentence structure-level perspectives, we have constructed triplets composed of anchor text, positive text, and negative text as augmented dataset. This dataset has been validated to significantly enhance the performance of pre-trained models. We intend to make this dataset publicly available.
- We propose an Enhanced Transformer-based Semantic Matching method, designed for few-shot learning through weakly contrastive pre-training. EBSIM employs a plug-and-play contrastive pre-training module during the training phase, based on relation-aspect and Alignment-Aspect, to optimize the bidirectional semantic understanding between texts.
- We conducted comprehensive experiments on five Chinese and English semantic matching datasets using multiple pre-trained language models. The experimental results confirm that our enhanced pre-training method significantly improves the model's performance on semantic matching tasks. Our source code and data will be publicly released.

## 2 RELATED WORK

### 2.1 Semantic Text Similarity Methods

Generally, deep semantic text similarity (STS) modeling includes two kinds of methods: interaction-based method and representation-based method. The former usually constructs feature interactions between two sentences, and further obtains the matching result [13, 19, 64], which obtain better performances with interaction mechanisms. The latter usually learns a representation and calculates the similarity score through a deep learning model [1, 9, 34].

In recent years, STS has rapidly developed with the help of pre-training models. Based on pre-trained BERT models, two kinds of fine-tuned architecture are typically used: bi-encoders and cross-encoders. A lot of work aims to balance these two methods in terms of efficiency and performance [24, 28, 41, 47]. Different from the above works, we focus on enhancing pre-training for STS tasks. We utilize the cross-encoders fine-tuning method, and this stage is identical to the fine-tuning of the conventional PLMs.

### 2.2 Pre-training Language Models

PLMs can capture rich language information from the text and benefit many applications by fine-tuning [45]. Previous PLMs focus on the general language model, most of which are following BERT [16]. BERT-wwm[15] adapts whole word masking in Chinese text, masking the whole word instead of masking Chinese characters. RoBERTa [38] is an improved version of BERT, including the use of dynamic masking and more training data, while removing the next sentence prediction.

More related to our work is the enhancement of pre-training for specific tasks. MWA [35] enhances pre-trained Chinese character representation with the word-aligned attention. SKEP [53] introduces sentiment knowledge enhanced pre-training to learn

a unified sentiment representation. To better model dialogue behaviors, TOD-BERT [58] proposes a contrastive objective function to simulate the response selection task. TaskPT [21] adds a task-guided pre-training stage with selective masking to capture the domain-specific and task-specific patterns. Similar but different to these works, we are the first to enhance pre-training for the STS tasks. There has not been the same job before.

## 2.3 Contrastive Learning

In recent years, contrastive learning plays an important role in self-supervised learning for computer vision, natural language processing, and other domains [26, 56, 65]. Contrast visual representation learning focuses on generating effective visual representations [23, 42]. Recent methods have produced results comparable to the state-of-the-art supervised method on some tasks [4, 11, 22]. Moreover, some previous works solve specific tasks through contrastive learning. LS-Score[59] evaluates the quality of abstracts through unsupervised contrastive learning. GCL[3] solves the neural dialogue generation with group-wise contrastive learning.

More related to our work is the language representation learning. Contrastive learning assumes that observed pairs of text are more semantically similar than randomly sampled text[2]. Some methods try to model the language representation learning with the idea of contrastive learning [12, 30, 60]. Unlike these previous works, we construct sentence pairs and design the contrast objectives to enhance the sentence similarity relation modeling.

## 3 ENHANCING PRE-TRAINING FOR SEMANTIC TEXT SIMILARITY

This section will describe the proposed method in detail. Section 3.1 formalizes STS task and the pipeline of our method, followed by weakly-supervised text similarity corpus construction in Section 3.2. Finally, contrastive learning objectives are described in Section 3.3.

### 3.1 Overview

In this section, we formalize the semantic text similarity (STS) task and describe the pipeline of EBSIM. Given a data set $\mathcal{D} = \{(s_{a1}, s_{b1}), (s_{a2}, s_{b2}), ..., (s_{an}, s_{bn})\}, (n \in N)$, we aim to learn a similarity model $\mathcal{M}$ from $\mathcal{D}$. Thus, $\mathcal{M}(s_a, s_b)$ measures the similarity degree between $s_a$ and $s_b$ for any new sentence pair $(s_a, s_b)$, where the similarity degree is a regression score or classification label. In BERT-based semantic text similarity measurement, we use the embedding of '[CLS]' tokens to predict similarity labels. The pipeline we propose for enhanced pre-training in semantic matching mainly consists of three stages, as outlined in Figure 1.

### 3.2 Large-scale Weakly-supervised Data Augmentation

As previously analyzed, there exists a natural gap between pre-training learning objectives like the masked language model, sentence order prediction, and semantic matching objectives. To enhance the consistency between the pre-training and fine-tuning stages, optimization is required from both the pre-training data and pre-training methods. A critical challenge is to design effective data augmentation and training methods that align with the

characteristics of semantic matching tasks. This allows models to capture semantic relationships between texts from the augmented data and accurately recognize semantics through enhanced training methods.

In particular, semantic similarity primarily focuses on the semantic connections between texts or concepts rather than surface similarity. Models in semantic similarity tasks can easily be misled by two types of similar schemes, namely token-level similarity and sentence structure-level similarity. Token-level similarity refers to situations where two texts share similar or overlapping words in their textual forms. In such cases, models may mistakenly assume that tokens with similar forms also share similar semantics. Sentence structure-level similarity pertains to situations where two texts have similar sentence structures or expressions. In these cases, models may incorrectly assume that texts with similar structures also convey similar semantics. Therefore, we design two data enhancement methods to construct weakly supervised data sets from the perspective of enhancing the model's ability to recognize these two schemes. Subsequently, we guide the model's learning through finely designed enhancement pre-training objectives that correspond to these two schemes.

#### 3.2.1 Token-level Similarity Scheme.
In order to construct samples with token-level similarity for enhancing pre-training, a cost-effective data augmentation method needs to be designed. Given that titles and sentence contents in chapter-level corpora simultaneously possess rich semantic and token similarity relationships, we build samples based on general pre-training corpora such as Wikipedia.

For each article, including the title $t$ and the content set of sentences $S$, we consider retrieving related sentences through its title. We encode the title $t$ and all sentences in content $\{s_1, s_2, ..., s_K\} \in S$ through a sentence encoder, where $K$ is the number of sentences in the current article content. Then we calculate the similarity and sort these sentence vectors $\{v_1, v_2, ..., v_K\} \in V$ using cosine similarity:

$$Sim(v_a, v_b)|_{cos} = \frac{v_a \cdot v_b}{\max\left(\|v_a\|_2 \cdot \|v_b\|_2, \epsilon\right)} \quad (1)$$

where $v_a$ and $v_b$ are sentence vectors, and $\epsilon$ is a small value to avoid division by zero. In our work, we obtain the sentence vector by computing the mean of all output vectors of PLMs.

After sorting, we can get relevant candidates. These relevant candidates can be combined with the title to form sentence pairs. The order of candidates represents the similarity order of sentence-pairs:

$$\begin{cases} < title, relevant_1 > \\ < title, relevant_2 > \\ ... \\ < title, relevant_K > \end{cases} \quad (2)$$

where $< title, relevant_1 >$ is more similar than $< title, relevant_x >$ $(1 < x \leq K)$. As a result, we can construct the weakly-supervised sentence triples $< title, relevant_1, relevant_x >$.

#### 3.2.2 Structure-level Similarity Scheme.
Considering the impact of similar sentence structures on semantic matching effectiveness, we construct augmented samples based on sentences with similar structures to aid model training. One simple and effective way is

to mask synonym entities in the text while preserving other tokens, thereby retaining similar sentence patterns and structures. Since the $< title, relevant_1 >$ pairs obtained earlier are of high quality, we directly perform data augmentation on such sample pairs. Specifically, we calculate the word similarity between the title and relevant1 based on the text representation vectors. Then, we mask words in $relevant_1$ with a similarity greater than a certain threshold, resulting in a new text $relevant_1^{(drop)}$, with similar sentence structures. The function for constructing similar sentence structures can be represented as follows:

$$s_{i,j}' = f_{drop}(s_{i,j}) = \begin{cases} s_{i,j} & \text{if } sim(s_{i,j}, *) < \eta, \\ [MASK] & \text{if } sim(s_{i,j}, *) \geq \eta. \end{cases} \quad (3)$$

where $s_{i,j}$ represents the $j$-th word in the sentence $s_i$, which is expressed as $relevant_1$ above. $s_{i,j}'$ denotes the generated word by structure similar function, which is expressed as $relevant_1^{(drop)}$ above. $[MASK]$ token represents a meaningless placeholder token, just like in other general pre-trained language models. $sim(s_{i,j}, *)$ denotes the similarity between the $j$-th word in $relevant_1$ and the word in $title$ calculated by cosine similarity. $\eta$ indicates the similarity threshold used as a hyperparameter to control the level of masking.

Based on sentence pair $< title, relevant_1 >$, we can use the similar sentence structure component method above to get the second kind of weakly supervised augmentation sample as follows:

$$< title, relevant_1, relevant_1^{(drop)} > \quad (4)$$

This kind of sentence triples makes the model face the challenge of similar sentence structure during training, so as to help the model recognize semantically similar information more accurately. It is worth mentioning that for $< title, relevant_1 >$, if there are no synonyms that exceed the threshold, no triples will be constructed. Therefore, for each document, 1-2 triples are constructed.

## 3.3 Multi-Aspects Contrastive Pre-training

Compared to the labeled high-quality similar sentences, the similarity labels of weakly-supervised sentence pairs above are relatively weak in reliability. As a result, it can't be used directly for optimizing the classification objective. Inspired by contrastive learning's success, we find the order of similarity is valuable and trustworthy. Therefore, we introduce the comparative prediction task, which is helpful for similarity relation learning.

Given the input sentence triple $< t, s_p, s_n >$, where $s_p$ is more similar to $t$ than $s_n$. We want to learn a classifying function with the ability to identify $s_p$ is more similar than $s_n$. A more common practice is Triplet Loss [49], which encourages a large margin between the similarity of the anchor $t$ and its positive $s_p$ and the similarity of the anchor and negative $s_n$. Mathematically, we minimize the following loss function:

$$\mathcal{L}(t, s_p, s_n, \xi, \beta) = \max \left( \xi(t, s_n) - \xi(t, s_p) + \beta, 0 \right) \quad (5)$$

where $\xi$ is the similarity measurement function. Margin $\beta$ ensures that $s_p$ is at least closer to $t$ with gap $\beta$ than $s_n$, and we set $\beta = 1$ in our experiments.

Consequently, our goal is to define the appropriate function $\xi$, which can better model the semantic similarity relation between sentences. Our proposed contrastive objectives are committed to multiple mapping functions.

In the process of BERT-based semantic text similarity, we first concatenate two sentences to the required format. '[SEP]' is adopted to concatenate two sentences, and '[CLS]' is added at beginning. In detail, we concatenate a sentence pair $< s_a, s_b >$ and fed them into the PLM. With the help of semantic knowledge of PLM, the hidden states are calculated as follows:

$$H_{s_a:s_b}, h_{cls} = \text{BERT}(s_a, s_b) \quad (6)$$

where $h_{cls}$ denotes the representation of the first token '[CLS]'. $H_{s_a:s_b}$ represents the sequence representation of the whole input. In this paper, we aim to make full use of $h_{cls}$ and $H_{s_a:s_b}$ to model the semantic similarity relation between $s_a$ and $s_b$ via multi-aspect contrast.

*3.3.1 Relation-Aspect Contrast.* The representation of token '[CLS]' is regarded as the global representation of semantic relations, which contains the current sentence relation information and serves as the input for the classification layer in the fine-tuning stage. As a result, our first object is to model the similarity of sentences in the relation aspect. Through a distance transformation matrix, we simulate to calculate the distance of sentence pairs:

$$\mathcal{D}(s_a, s_b) = \sigma(W_d \cdot h_{cls} + b_d) \quad (7)$$

where $W_d, b_d$ are learnable parameters and $\sigma$ is the sigmoid function. Finally, we redefine the mapping function as the similarity function of the triple loss function:

$$\xi_{RA}(s_a, s_b) = -\mathcal{D}(s_a, s_b) \quad (8)$$

The Relation-Aspect (RA) contrastive objective is:

$$\mathcal{L}_{RA} = \mathcal{L}(t, s_p, s_n, \xi_{RA}) \quad (9)$$

*3.3.2 Alignment-Aspect Contrast.* The relation-aspect contrast focuses on the relationship between sentence pairs based on BERTs with multi-head self-attention. However, for each concatenated sentence pair, the global relation-aspect encoding relies mainly on the whole tokens' self-attention mechanism. There lacks of attention to local sentence alignment, which leads to insufficient interaction between the two sentences. A simple form of alignment based on the attention mechanism is used following [61] with minor modification to help the BERT-based STS. In this subsection, we leverage the alignment mechanism and propose an alignment-aspect contrast strategy to enhance interaction between sentences.

The alignment component takes features from $H_{s_a:s_b}$ as input and computes the aligned representations as output. Input from the first sentence $s_a$ of length $l_a$ is denoted as $s_a = (a_1, a_2, ..., a_{l_a})$ and input from the second sentence $s_b$ of length $l_b$ is denoted as $s_b = (b_1, b_2, ..., b_{l_b})$. The output representation $a'$ and $b'$ are computed by weighted summation of representations. The summation is weighted by attention scores between the current position and

the corresponding positions in another sentence:

$$a'_i = \sum_{j=1}^{l_b} \frac{\exp(a_i \cdot b_j)}{\sum_{k=1}^{l_b} \exp(a_i \cdot b_k)} b_j$$
$$b'_j = \sum_{i=1}^{l_a} \frac{\exp(a_i \cdot b_j)}{\sum_{k=1}^{l_a} \exp(a_k \cdot b_j)} a_i \qquad (10)$$

The fusion layer compares raw token and aligned representations from two perspectives ($\cdot$ and $-$). The fusion result of sentence $s_a$ is computed by:

$$a_i^f = W_f * [a_i; a'_i; a_i \circ a'_i; a_i - a'_i] + b_f \qquad (11)$$

where $W_f, b_f$ are learnable parameters, and $\circ$ denotes element-wise multiplication. The subtraction operator highlights the difference between the two vectors while the multiplication highlights similarity. Then, we obtain $s_a^f = \text{MaxPooling}(A^f)$ and $s_b^f$ omitted here. We simulate to calculate the similarity confidence of sentence pairs based on the alignment mechanism:

$$\mathcal{A}(s_a, s_b) = \sigma(W_{\mathcal{A}} \cdot [s_a^f \circ s_b^f; s_a^f - s_b^f] + b_{\mathcal{A}}) \qquad (12)$$

The Alignment-Aspect (AA) contrastive objective is:

$$\xi_{AA}(s_a, s_b) = \mathcal{A}(s_a, s_b)$$
$$\mathcal{L}_{AA} = \mathcal{L}(t, s_p, s_n, \xi_{AA}) \qquad (13)$$

The overall objective function is:

$$\mathcal{L} = \mathcal{L}_{RA} + \mathcal{L}_{AA} \qquad (14)$$

## 3.4 Semantic Matching Fine-tuning

In the fine-tuning phase, our primary focus is on training and testing the model for semantic text matching tasks. To simplify, we consider the semantic text matching task as determining whether two texts have similar semantic information, which can be defined as a classification problem. If two texts are semantically similar, the corresponding label is set to 1. If two texts do not share semantic similarity, the corresponding label is set to 0. The model's task is to predict the probability of semantic similarity between the two input texts.

Based on the encoded [CLS] representation $h_{cls}$, we can obtain the semantic similarity probability as:

$$r_i = softmax(MLP(h_{cls})) \qquad (15)$$

where $g$ represents the similarity function, which is replaced by the inner product. Let $y_i \in \{0, 1\}$ represent the real label of the sample. The cross entropy loss function can be expressed as:

$$L = -\frac{1}{N} \sum_{i=1}^{n} [y_i log r_i + (1 - y_i) log(1 - r_i)] \qquad (16)$$

where $y_i$ represents the label of the sample. $N$ denotes the number of training samples.

## 4 EXPERIMENTS

In this section, we conduct extensive experiments to answer the following questions:

- **RQ1** What kind of effect does our proposed EBSIM enhancement pretraining framework have on pretrained language models in semantic matching tasks?
- **RQ2** Can our proposed EBSIM improve the effectiveness of few shot learning on downstream semantic matching tasks?
- **RQ3** What impact do the designed contrastive pre-training modules based on relation-aspect and alignment-aspect have on the model effect?
- **RQ4** Does our constructed weakly supervised samples help to improve the effect of the pre-trained model? What is the effect of different training sample scales on the model effect?
- **RQ5** Can the training method based on alignment-aspect contrast learning objective help the model improve the accuracy of semantic recognition?

### 4.1 Experimental Settings

*4.1.1 Datasets.* To verify the effectiveness of our EBSIM, we conduct experiments on Chinese and English corpus respectively. We collect five different STS datasets. The dataset statistics are shown in Table 1. **BQ** is the largest manually annotated Chinese public corpus in the bank domain [7]. **LCQMC** is a large-scale Chinese question matching corpus [36]. Followed with Chen et al. [7], we mainly report Accuracy scores for BQ and LCQMC. **MRPC** is a corpus of sentence pairs automatically extracted from online news sources [17]. **QQP** is a collection of question pairs from the community question-answering website Quora[1]. **STS-B** consists of English datasets used in the semantic textual similarity tasks, which are organized in the context of SemEval between 2012 and 2017 [5]. We follow common practice and report both Pearson and Spearman scores. Following Wang et al. [57], we use the training set, validation set and test set corresponding to the dataset. Because the classes in MRPC(68% positive) and QQP(37% negative) are imbalanced, we follow common practice and report both Accuracy and F1 scores.

*4.1.2 Implementation Details.* All the experiments are executed on the base model (12 layers, hidden size 768), including BERT-base, BERT-wwm-ext-base, RoBERTa-base, and so on. All the experiments are tested on five random seeds. We report the average score to evaluate the performance of these models.

**Corpus Construction.** During constructing the weakly-supervised corpus, we use $\text{BERT}_{base}$ as a encoder to obtain the sentence embedding. The document-level corpus include English corpus [55] and Chinese corpus[2]. We construct weakly-supervised sentence triples $< title, relevant_1, relevant_2 >$ as described in Section 3.2. During the synonyms dropping, the embedding size is set to 200. For Chinese corpus, we initialize the Chinese word embedding from Tencent Embedding Corpus[3]. For English corpus, word2vec [39] is initialized randomly and fine-tuned on our English corpus. The similarity threshold for word pairs to be masked is 0.8. The number of masked words does not exceed 50% of the entire sentence. For all experiments, the number of documents is 100w, except Table 4 uses different scales (200,000 and 500,000). After the weakly supervised dataset construction as described in 3.2, the numbers of final

---

[1]https://www.quora.com/q/quoradata/First-Quora-Dataset-Release-Question-Pairs
[2]https://github.com/brightmart/nlp_chinese_corpus
[3]https://ai.tencent.com/ailab/nlp/en/index.html

| Dataset | MaxLen | Batch Size | Epoch | Learning rate | Train | Dev | Test | Language |
|---------|--------|------------|-------|---------------|-------|-----|------|----------|
| BQ | 128 | 64 | 3 | 3e-5 | 100k | 10k | 10k | Chinese |
| LCQMC | 128 | 64 | 3 | 2e-5 | 240k | 8.8k | 12.5k | Chinese |
| MRPC | 128 | 32 | 3 | 5e-5 | 3.7k | 408 | 1.7k | English |
| QQP | 128 | 32 | 3 | 5e-5 | 390k | 5k | 5k | English |
| STS-B | 128 | 32 | 3 | 5e-5 | 5.7k | 1.5k | 1.4k | English |

Table 1: Benchmarks Settings. The Train, Dev and Test denote the size of corresponding dataset respectively. k represents the numerical value one thousand.

| Model | MRPC | | QQP | | STS-B | |
|-------|------|------|------|------|-------|------|
| | Acc | F1 | Acc | F1 | Pearson | Spearman |
| **Previous Systems** | | | | | | |
| ERNIE2.0 [51] | 86.1 | 89.9 | - | - | 87.6 | 86.5 |
| R$^2$-Net [63] | 84.3 | - | **91.6** | - | - | - |
| tBERT [44] | - | 88.4 | - | 90.5 | - | - |
| **Re-Implementation** | | | | | | |
| DistilBERT$_{base}$[48] | 82.72 | 87.67 | 90.09 | 90.04 | 82.67 | 81.62 |
| XLNET [62] | 85.28 | 89.08 | 90.95 | 90.87 | 86.04 | 85.24 |
| ALBERT$_{base}$ [32] | **86.55** | 89.98 | 91.20 | **91.10** | 87.92 | **86.85** |
| **Our Implementation** | | | | | | |
| BERT$_{base}$ [16] | 84.70 | 88.64 | 90.86 | 90.71 | 85.44 | 84.31 |
| + EBSIM | **85.39** | **89.47** | **91.42** | **91.38** | **87.12** | **86.24** |
| RoBERTa$_{base}$ [38] | 87.36 | 90.63 | 91.02 | 90.95 | 88.84 | 88.11 |
| + EBSIM | **87.65** | **90.81** | **91.41** | **91.35** | **89.42** | **88.76** |

Table 2: The Accuracy(%) and F1(%) for semantic text similarity on the MRPC, QQP and STS-B datasets.

triples: 1,750,000(English), 1,750,000(Chinese). These final triples include 1,000,000 triples constructed through retrieval and others constructed by synonyms deletion.

**Pre-Training and Fine-Tuning Configuration.** The pre-training is based on the initialization parameters of existing PLMs such as BERT, RoBERTa. During further pre-training, We use Adam optimizer[29]. The learning rate is set to 5e-6, and the batch size is 32. We choose the best pre-trained model for the fine-tuning according to the dev set, including 20,000 samples. This fine-tuning stage remains unchanged. For the hyper-parameters of fine-tuning, we refer to the followed existing works [15, 16, 37, 51] and explore the most suitable ones. The details of fine-tuning for benchmarks are as Table 1.

*4.1.3 Baselines.* To evaluate the effectiveness of our approach, we compare the model with the following baselines: ERNIE2.0 [51], R$^2$-Net [63], tBERT [44], DistilBERT[48], XLNET [62], ALBERT [32], BiMPM[57], BERT-wwm [15], BERT-wwm-ext [15], RoBERTa-wwm-ext [15], MacBERT [14], ERNIE [50], ERNIE2.0 [51], GMN-BERT[8], BERT[16], BERT-wwm-ext [15], RoBERTa [38].

## 4.2   Overall Performance (RQ1)

In order to achieve fair and objective performance evaluation, we conducted enhanced pretraining experiments on three representative Transformer-based models, namely BERT, BERT-wwm-ext [15] , and RoBERTa [38], on the aforementioned five Chinese and English datasets. The experimental results for the Chinese datasets are presented in Table 2, while the results for the English datasets

can be found in Table 3. Based on the experimental comparisons, we made the following observations:

- For the Chinese semantic matching tasks, BQ and LCQMC, our proposed enhanced pretraining method, EBSIM, resulted in significant performance improvements. Specifically, for the BQ dataset, the enhanced BERT-base model achieved an accuracy improvement of 0.5%, the BERT-wwm-ext model improved by 0.69%, and the RoBERTa model improved by 0.93%. Due to the relatively large sample size of the LCQMC dataset, the improvement range for the three pretrained language models ranged from 0.36% to 0.63%. The experimental results effectively confirm the effectiveness of our weakly supervised contrastive enhancement pretraining method. It's worth noting that our weakly supervised dataset was constructed based on the original training data of these pre-trained language models.

- For the English semantic matching tasks, MRPC, QQP, and STS-B, our method also demonstrated superior performance. Specifically, for the high-quality STS-B dataset, the enhanced BERT-base model achieved a 1.68% improvement in Pearson correlation, while the enhanced RoBERTa model improved by 0.58%. For the MRPC and QQP datasets, the accuracy improvement of the enhanced pretrained language models ranged from 0.22% to 0.69%. Whether in tasks with larger sample sizes or smaller semantic matching tasks, our enhanced pretraining strategy provided noticeable benefits to the models.

| Model | BQ | LCQMC |
|---|---|---|
| **Previous Systems** | | |
| BiMPM[57] | 81.9 | 83.4 |
| BERT-wwm [15] | 85.2 | 87.0 |
| BERT-wwm-ext [15] | 85.3 | 87.1 |
| RoBERTa-wwm-ext [15] | 85.0 | 86.4 |
| MacBERT [14] | 85.2 | 87.0 |
| ERNIE [50] | 84.8 | 87.4 |
| ERNIE2.0 [51] | 85.0 | **87.9** |
| GMN-BERT[8] | **85.6** | 87.3 |
| **Our Implementation** | | |
| BERT$_{base}$ [16] | 84.87 | 87.05 |
| + EBSIM | **85.37** | **87.47** |
| BERT-wwm-ext [15] | 84.95 | 87.02 |
| + EBSIM | **85.64** | **87.38** |
| RoBERTa$_{base}$ [38] | 84.42 | 86.59 |
| + EBSIM | **85.35** | **87.22** |

Table 3: The Accuracy(%) for semantic text similarity on the BQ and LCQMC datasets.

## 4.3 Few-shot Learning Performance (RQ2)

One of the research objectives of this study is to enhance pretrained language models for semantic matching, so that they can maintain strong representational capabilities in the downstream few-shot learning scenario, where the training samples are limited. For the QQP dataset with 390K training samples, we conducted experiments with subsets of the original training set containing 20%, 40%, 60%, 80%, and 100% of the samples for fine-tuning, followed by testing the model's performance on the original test set. The model's performance under different proportions of training samples is presented in Figure 2. Additionally, to investigate whether our proposed method also helps with the convergence of models on semantic matching tasks, we calculated the model's performance on the test set based on different training steps, as shown in Figure 3.

- Visually, it can be observed that our method not only outperforms the original BERT model when using 100% of the training samples but also maintains a stable advantage even when using only 60% or 20% of the training samples. This indicates that our proposed enhanced pretraining method indeed effectively aids the model in accurately recognizing semantic similarity, regardless of the scale of the training sample.
- It is evident that our proposed method achieves significantly higher performance when using a limited number of training samples. Particularly, the model enhanced by EBSIM achieves better performance with only 60% of the training samples compared to the original pretrained model using 100% of the training samples. In other words, our proposed method can directly improve the few-shot learning performance on consistent downstream tasks solely through effective enhancement in the pretraining phase.
- Based on the test results corresponding to different training steps shown in Figure 3, EBSIM accelerates the model's convergence during the fine-tuning phase. In the initial 5000

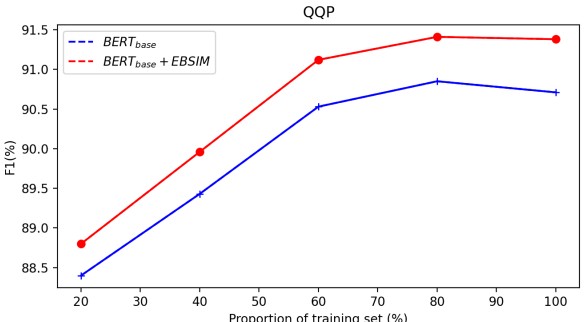

Figure 2: The F1 scores of different QQP training set scales in the fine-tuning stage. The EBSIM's improvement are stable with different training set scales.

steps, the model enhanced by EBSIM already exhibits significantly better performance than the original pretrained language model.

## 4.4 Ablation Experimental Study (RQ3)

In order to validate the effectiveness of our contrastive pretraining optimization objectives based on relation-aspect (RA) and Alignment-Aspect (AA), we conducted ablation experiments. The experimental results are presented in Table 5, where we removed each of the two contrastive learning objectives separately. The results show a significant drop in performance on the BQ and MRPC datasets when we removed both contrastive learning objectives. This indicates that our adopted weakly supervised training method can indeed help the model improve its judgment of semantic similarity. Further observation reveals that RA has a greater impact on the model's performance compared to AA. This is because RA primarily utilizes global multi-head attention to perceive semantic tokens, which inherently contains global semantic information between the two texts. On the other hand, the introduction of AA mainly helps the model align more fine-grained token information between the two texts, compensating for the insufficient perception of some local tokens caused by global attention. In fact, since our weakly supervised samples were constructed while considering the characteristics of semantic matching tasks, our weakly supervised training objectives capture the modeling approach for semantic similarity. Therefore, the combination of these two objectives has a significant effect.

## 4.5 Pre-training Sample Scale Analysis (RQ4)

The goal of this paper is to explore a low-cost training strategy for enhancing existing pretrained language models. In order to gain a more detailed understanding of the impact of enhanced pretrained samples on model performance, we conducted an in-depth analysis of the effect of pretrained sample size. Specifically, we performed model-enhanced pretraining using weakly supervised samples of 200,000, 500,000, and 1,000,000. We then tested the performance of models with different pretrained sample sizes on downstream semantic matching tasks. For convenience, we used BERT-base as the pretrained language model for validation. The experimental results are shown in Table 4. It can be observed that as the number of

| Model | BQ | | LCQMC | | MRPC | | QQP | | STS-B | |
|---|---|---|---|---|---|---|---|---|---|---|
| | Acc | F1 | Acc | F1 | Acc | F1 | Acc | F1 | Pearson | Spearman |
| $BERT_{base}$ | 84.87 | 84.77 | 87.05 | 87.97 | 84.70 | 88.64 | 90.86 | 90.71 | 85.44 | 84.31 |
| $BERT_{base}$ + EBSIM(20w) | 85.10 | 84.85 | 87.14 | 88.07 | 84.93 | 89.15 | 91.15 | 91.03 | 86.41 | 85.20 |
| $BERT_{base}$ + EBSIM(50w) | **85.49** | **85.35** | 87.34 | 88.17 | 85.10 | 89.22 | 91.09 | 90.99 | 86.68 | 85.58 |
| $BERT_{base}$ + EBSIM(100w) | 85.37 | 85.18 | **87.47** | **88.28** | **85.39** | **89.47** | **91.42** | **91.38** | **87.12** | **86.24** |

**Table 4: The experimental results with different scale datasets in the pre-training stage. 20w, 50w, 100w refer to the number of documents. For each document, 1-2 triples are constructed as described in Section 3.2. w represents the numerical value ten thousand.**

| Model | BQ | | MRPC | |
|---|---|---|---|---|
| | Acc | F1 | Acc | F1 |
| $BERT_{base}$ + EBSIM | 85.37 | 85.18 | 85.39 | 89.47 |
| −AA | 85.06↓ | 84.95↓ | 85.10↓ | 89.07↓ |
| −AA − RA | 84.87↓ | 84.77↓ | 84.70↓ | 88.64↓ |

**Table 5: The experimental results of different ablation strategies. AA and RA refers to the pre-training objectives: Alignment-Aspect and Relation-Aspect.**

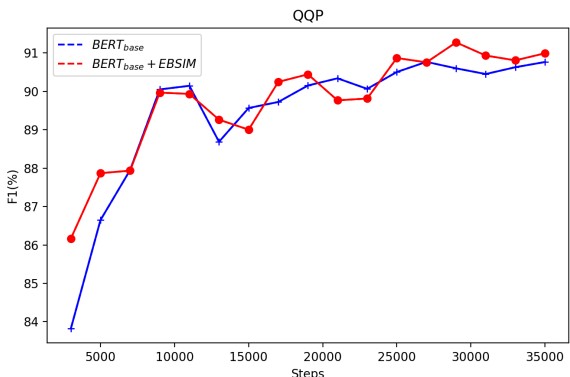

**Figure 3: The F1 scores of BERT and EBSIM on QQP dev set when trained for up to 35000 steps. In fewer training steps, the EBSIM can also achieve better results than BERT.**

weakly supervised samples used for enhanced pretraining increases, the model's performance shows an upward trend. When training with the complete 1,000,000 samples, the model achieves optimal performance on both datasets. The experimental results validate the effectiveness of the weakly supervised samples we constructed and demonstrate that this type of sample can consistently improve the model's performance as the quantity increases.

## 4.6 Alignment Mechanism Analysis (RQ5)

The main purpose of introducing the Alignment-Aspect in the contrastive pretraining optimization objective is to achieve fine-grained semantic alignment between tokens in the text, thereby assisting the model in accurate semantic matching recognition. We aim to optimize the model's semantic perception of certain keywords more effectively. Therefore, we conducted a case study

to further explore the enhanced model's capabilities. By visualizing the distribution of attention weights among different models in Figure 4, we can observe that by introducing the alignment-aspect contrastive mechanism, our model can focus more on key words in sentences, which may be the reason why the alignment-aspect contrastive mechanism improves the performance. In this case, in order to judge the semantic similarity of two sentences: *How do I quit smoking* and *How do I give up on cigarette smoking*, EBSIM pays more attention to *quit smoking* and *give up on cigarette smoking* than BERT. They are keywords to identify the similarity.

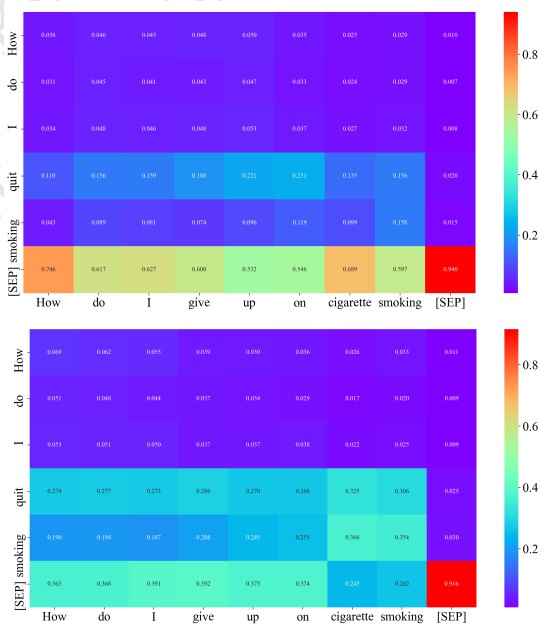

**Figure 4: The distribution of alignment attention weights between BERT (Up) and EBSIM (Down).**

## 5 CONCLUSION

In this paper, we propose a contrastive pre-training method to enhance Transformer-based semantic text similarity called EBSIM on the weakly-supervised corpus, which fills the gap between the pre-training and the fine-tuning in this area. Plenty of comparative experiments show the effectiveness of the proposed strategy. In the future, we will investigate the possibility that the enhanced mechanism is applied on other tasks.

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
