# OpenReview forum: "Enhancing Transformer-based Semantic Matching for Few-shot Learning through Weakly Contrastive Pre-training"
_acmmm.org/ACMMM/2024/Conference — MM2024 Poster_

### Official Review · Reviewer_fKj5 · 2024-05-24

**Rating:** 5
**Confidence:** 3

**Summary:**

The paper presents an innovative approach to enhancing Transformer-based semantic text matching for few-shot learning scenarios through a method named EBSIM (Enhanced Transformer-based Semantic Matching for Few-shot Learning through Weakly Contrastive Pre-training). The authors focus on addressing the challenges of data quality and quantity in practical downstream applications and propose a general enhancement scheme for few-shot semantic text matching tasks. EBSIM introduces a weakly supervised data augmentation method and a contrastive learning objective to improve the semantic similarity measurement between texts. The method was evaluated on five Chinese and English semantic text matching datasets using various Transformer-based pre-trained models, demonstrating significant improvements in performance.

**Strengths:**

(1) Innovative Approach: The paper proposes a novel pre-training strategy that bridges the gap between pre-training and fine-tuning stages for semantic text matching tasks.

(2) Weakly Supervised Data Augmentation: A cost-effective method for creating weakly supervised samples that enhances the model's ability to understand semantic similarity.

(3) Contrastive Learning Objective: The introduction of a contrastive learning objective based on alignment aspects leads to effective semantic matching by optimizing bidirectional semantic perception.

(4) Comprehensive Experiments: The method was tested on multiple datasets in both Chinese and English, providing a robust evaluation of its effectiveness.

(5) Performance Improvement: The results show significant performance improvements over existing models on semantic text matching tasks.

(6) Ablation Studies: The paper includes ablation experiments that validate the contribution of different components of the proposed method.

**Limitations:**

(1) Limited Generalizability: The enhancements are specifically tailored for semantic text matching tasks and may not be directly applicable to other types of NLP tasks.

(2) Resource Intensive: Although the data augmentation method is cost-effective, the pre-training process with contrastive learning objectives might still require considerable computational resources.

(3) Overreliance on Synonyms: The method might still be sensitive to the presence of synonyms, which could affect the accuracy of semantic judgments.

(4) Dataset Bias: The effectiveness of EBSIM was evaluated on specific datasets, and its performance on other types of datasets or domains might vary.

(5) Complexity of Implementation: The introduction of new objectives and mechanisms may increase the complexity of the model, potentially making it harder to implement and maintain.

(6) Potential for Overfitting: With the focus on enhancing performance on specific tasks, there may be a risk of overfitting to the particularities of the training datasets used.

**Suitability:**

2

---

### Official Review · Reviewer_RLbt · 2024-05-24

**Rating:** 4
**Confidence:** 2

**Summary:**

This paper proposes addressing the quality and quantity issues of downstream data when fine-tuning pre-trained models. Focusing on the task of semantic matching, the authors suggest using few-shot learning through weakly contrastive pre-training. Comprehensive experiments on five Chinese and English semantic text-matching datasets demonstrate the effectiveness of the proposed method.

**Strengths:**

1. The paper proposes weakly-supervised data augmentation and contrastive pre-training for Transformer-based semantic matching.

2. The proposed method achieves state-of-the-art performance on five Chinese and English semantic text matching datasets.

**Limitations:**

In Figure 3, why does the EBSIM method accelerate the model's convergence compared to BERT base (blue line)? A detailed explanation and analysis are needed.

**Suitability:**

2

---

### Official Review · Reviewer_wo8H · 2024-05-24

**Rating:** 2
**Confidence:** 3

**Summary:**

The authors introduce a pre-training method (called EBSIM) for enhancing performance on downstream semantic text matching when only a few fine-tuning examples are available. In particular, at pre-training time, they enforce token-similarity and structure similarity through two triplet loss objectives, creating ad-hoc triplets through a corpus of English and Chinese sentences composed of titles and relevant document sentences. The authors show the effectiveness of their pre-training method by fine-tuning the pre-trained model on downstream semantic matching datasets, showing some improvements over methods pre-trained using standard BERT objectives (MLM). The authors also perform some ablation studies to show that the novel model is more sample efficient when fine-tuned.

**Strengths:**

- The paper is well-written and quite easy to follow.
- The motivation is clear and the introduction helps to clearly present the problem.
- The experiments have been conducted on sentence corpora spanning two very different languages (English, Chinese).

**Limitations:**

- Despite the reasonable motivation and design, the proposed method seems not achieving a remarkable margin improvement (below 1%) over the standard MLM pre-training method. It is true that, from the ablation study, the model seems easier to fine-tune, but, even from Figure 3, it seems that there is not a significant statistical difference between BERT and BERT + EBSIM.
- The idea of pre-training large transformer models is that a single pre-training schema may benefit many downstream tasks. However, in this work, pre-training has been designed to benefit only the downstream semantic matching task. To be more solid, I would have expected a different experimental design, in which the EBSIM pre-training method is then fine-tuned on other downstream tasks. Otherwise, it may be that EBSIM helps semantic matching tasks while worsening performance, for example, on text summarization or next-sentence prediction.
- Given that pre-training and fine-tuning are very much correlated in the proposed approach, I would have expected in the ablation studies also an experiment where the pre-training data was mixed with the downstream semantic matching data, instead of being used for pre-training. This would have evaluated how much the overall pre-training schema - which includes specific token-level and structure-level objectives - helped over the data themselves. It could be that, in the end, the whole method works a few decimal percentage points better only because more data is being used.
- Given that the authors employ BERT pre-trained on MLM as a starting base, I would have expected the same pre-training EBSIM schema to be used on other pre-trained models (e.g., RoBERTa), to evaluate how much the model-specific pre-training strategies influence the whole pipeline.

Minor comments:
- In the related works, the authors can consider citing some pre-trained vision-language models like VinVL [1] or VL-BERT [2], in which semantic image-text matching is the key pre-training objective.
- A more in-detail figure showing the architecture of the pre-training schema would have better summarized the proposed architecture.
- In Section 4.2, line 636-637, Chinese datasets should be in Table 3 and English in Table 2. Currently, they are inverted.

Considering that this paper only concerns unimodal scenarios (text), the weaknesses of the experimental results, and the potential design flaws, I lean towards the rejection of this paper at ACM MM.

[1] Zhang, Pengchuan, et al. "Vinvl: Revisiting visual representations in vision-language models." Proceedings of the IEEE/CVF conference on computer vision and pattern recognition. 2021.

[2] Su, Weijie, et al. "VL-BERT: Pre-training of Generic Visual-Linguistic Representations." International Conference on Learning Representations. 2019.

**Suitability:**

1

---

### Meta-Review · Area_Chair_ARkk · 2024-07-02

**Recommendation:** Accept (Poster)
**Confidence:** 3

**Metareview:**

The reviewers did not reach a consensus regarding this paper. The authors' rebuttal did not address all the issues raised. In particular, there are still concerns related to generalization (mentioned by two reviewers) and the overall relevance of the paper for the conference (the approach is unimodal).
In the end, the paper is borderline even after the reviewer went carefully through both the paper and the rebuttal. The small tendencies in the final rating toward acceptance make me suggest this decision.